# On the Shoulders of a Giant: Contributions of Thomas Grogan, MD to Hematopathology

**Yasodha Natkunam * and Roger A. Warnke**

Department of Pathology, Stanford University School of Medicine, Stanford, CA 94305, USA; rwarnke@stanford.edu
* Correspondence: yaso@stanford.edu; Tel.: +1-650-725-9354

**Abstract:** The story of Thomas Grogan, MD is one of the most compelling narratives in the modern history of pathology. Progressing from a quintessential academic pathologist to an entrepreneur and a renowned inventor, his remarkable journey is one of creativity, courage, and a keen focus on improving the care of cancer patients. By enabling precision health and empowering the pathologist in that mission, he transformed the landscape of diagnostic pathology. In this review, we describe some of his salient contributions and how his vision has shaped and continues to shape hematopathology today.

**Keywords:** hematopathology; immunohistochemistry; lymphoma; antibodies; automation

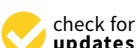

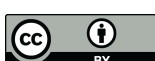

## 1. Introduction

In the Summer of 1978, a curious young physician with an interest in immunology and pathology arrived at Stanford University Medical Center. He had just completed two years as Chief of the Hematology Laboratory at the Walter-Reed Army Medical Center, Bethesda, Maryland, under the mentorship of Dr. Costan Berard, who was the Chief of Hematopathology at the National Institutes of Health [1]. His mentor assisted him in securing a fellowship to work with Drs. Ronald Dorfman and Roger Warnke at Stanford. Over the next three decades, he was to become a colossus who not only had the vision but the audacity to change the field of pathology.

Thomas Grogan, MD, had anything but a direct career path to hematopathology. His early life experiences, including serving in a U.S. National Institutes of Health (NIH) leprosy program on the Philippine island of Cebu, strongly influenced his trajectory in wanting to pursue a career in pathology [1]. At Walter-Reed Army Medical Center, he was exposed to the clinical and operational aspects of a hematology laboratory, which spurred him to explore hematopathology as a subspecialty. His arrival at Stanford coincided with the pioneering days of applying monoclonal antibodies to human tissue biopsies to aid in pathology diagnoses [2–4]. In this burgeoning field, he found his niche, where his interests in immunology and pathology merged and came to fruition.

After the completion of his fellowship, Dr. Grogan left Stanford to become an assistant professor at the University of Arizona where his vision of automation and precision diagnostics led him to build his own particular brand of entrepreneurship. He pioneered a fully automated immunohistochemistry platform, and in 1985, founded Ventana Medical Systems, Inc. Automated immunohistochemistry had a decisive impact on tissue-based diagnostics in scale and scope. Its rapid adoption was no surprise given the growing need for diagnostic tests that provided improved accuracy and turnaround time. As monoclonal antibody reagents expanded and technical aspects such as antigen retrieval methods enhanced assay quality, immunohistochemistry became a cornerstone of anatomic pathology. It provided standardization of assays and enabled the measurement of prognostic and therapeutic targets. These aspects were a significant boost for routine clinical workflows as well as for clinical trials and correlative studies across national and international medical

institutions. The standardization provided by automated immunohistochemistry was transformative in the field. It aided disease discovery and subtype classification and led the way for an integrated classification of hematopoietic neoplasms.

Thirty-five years later, immunohistochemistry still remains the main ancillary diagnostic test used in anatomic pathology practice. Flow cytometry, another method of immunophenotyping, which developed over the same period of time, provided the possibility of quantification in hematologic diagnoses; however, it did not offer the ability to incorporate spatial tissue architecture, which is key to anatomic pathology diagnoses. Similarly, newer cytogenetic and molecular technologies are also frequently used in hematopathology today; however, none have been as widely adopted for anatomic pathology diagnoses as automated immunohistochemistry.

Dr. Grogan's forays into technology and building a business while balancing an academic career and family, were not always easy or successful [1]. Despite many hurdles, his undeterred focus on improving patient care and his capacity for innovation sets him apart as a truly visionary leader. In the following sections, we describe some of his salient contributions that have and continue to enrich both diagnosis and research in hematopathology today.

## 2. Arrival at Stanford and Entry into Hematopathology

The connection between Drs. Costan Berard and Ronald Dorfman was instrumental in Dr. Grogan choosing Stanford for his training in hematopathology (Figure 1A). In the late 1960s and early 1970s, Dr. Berard was collaborating with oncologist Dr. Vince DeVita in pioneering studies in the treatment of Hodgkin and non-Hodgkin lymphomas at the NIH while Dr. Dorfman was collaborating with Drs. Henry Kaplan and Saul Rosenburg in similar studies at Stanford [5]. Drs. Berard and Dorfman saw each other often at the United States and Canadian Academy of Pathology (USCAP) as well as at other national and international lymphoma meetings. They traveled and taught together, including in a cross-country teaching adventure in Australia. Drs. Berard and Dorfman also shared a love of golf and their time spent golfing together often translated into impactful ideas that shaped the future of hematopathology. While traveling in Italy, they discussed the importance of a forum for sharing diagnostic and research advances in hematopathology and founded the Society for Hematopathology in 1981. This society still remains the premiere professional society for hematopathologists in the United States and Canada [6].

It was no wonder that at Dr. Berard's recommendation, Dr. Dorfman was especially eager to accept a highly qualified Dr. Grogan to train with him. There was, however, a small problem in that there was no hematopathology fellowship at Stanford at the time. To obtain funding for a one-year fellowship that combined clinical work and research was no small feat. After more than one failed attempt to procure external fellowship funding, a creative solution was found. Dr. Grogan was brought to Stanford as a junior faculty member for six months with sign-out responsibilities and a research post-doctoral fellowship for the remaining six months. Thus, Dr. Grogan spent his year at Stanford exposed to a large number of lymphoma cases on the clinical side and engaging in research in the Warnke laboratory. His clinical experience at Stanford involved spending a significant amount of time at the microscope with Dr. Dorfman in addition to his sign-out duties. On the research side, there were a plethora of new discoveries of monoclonal antibodies and immunophenotyping that kept him amply occupied. His engagement with the academic mission was immediate and profound. His tenure at Stanford proved enormously successful and foundational for the remainder of his career. In addition, Drs Grogan and Warnke became lifelong colleagues, collaborators and friends (Figure 1B).

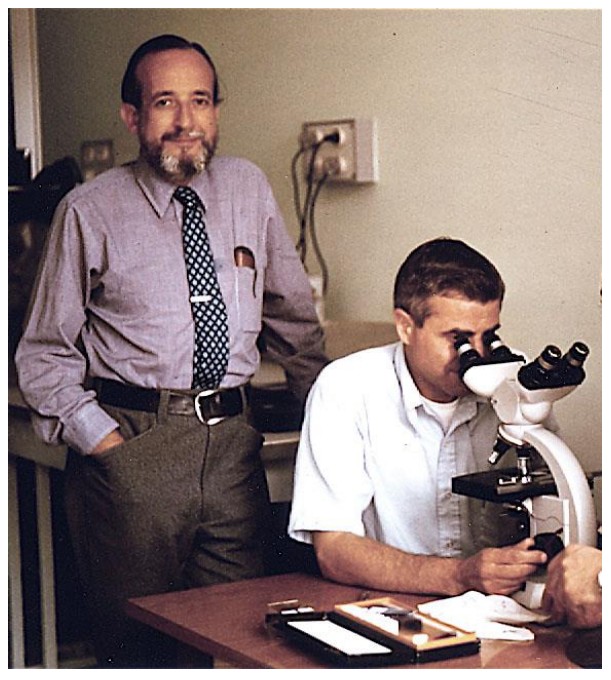

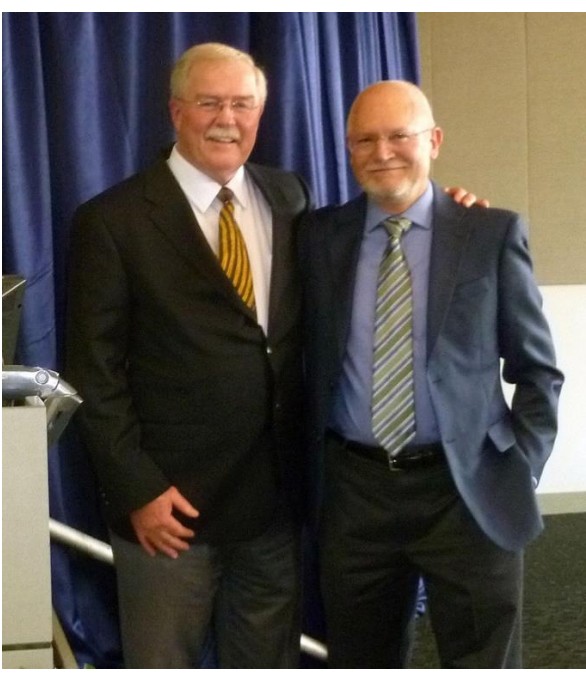

(A)                                                                                    (B)

**Figure 1.** (**A**) Photograph of Dr. Grogan's mentors, Drs. Ronald Dorfman and Costan Berard (at microscope) discussing a challenging case (1976). (**B**) Drs. Thomas Grogan and Roger Warnke, at the inaugural Warnke Lectureship delivered by Dr. Grogan at the Department of Pathology, Stanford University School of Medicine (2015).

## 3. Beginnings of Immunohistochemistry

An important pre-requisite for immunophenotyping hematopoietic neoplasms was the invention, in 1975, of hybridoma technology, which enabled the production of large numbers of monoclonal antibodies [7]. These antibodies were rapidly exploited for the purposes of developing diagnostics and therapeutics. From the late 1970s to the early 1980s, the Warnke laboratory began putting monoclonal antibodies, many produced in the laboratory of Dr. Ronal Levy, on frozen sections and cytospins of lymphomas [2–4,8–15]. Initially, Fab2 fragments of goat anti-mouse antibodies were used on frozen sections [2–4]. Dr. Levy had contributed many wonderful reagents from Israel where he had been a post-doctoral fellow at the Weizmann Institute of Science. Among them were immunoglobulin light chain antibodies conjugated to fluorochromes: anti-kappa was fluoresceinated and anti-lambda was rhodaminated. Under a fluorescent microscope, polyclonal B-cell proliferations showed a mix of green and red labeled B-cells whereas monoclonal B-cell proliferations showed a marked predominance of green or red labeled cells. During that time, the Howard Hughes Institute had an unexpected surplus of research funds at the end of a fiscal year, which allowed the purchase of a state-of-the-art fluorescent microscope. The apt combination of reagents and instrumentation formed the perfect ingredients for pioneering discovery in the Warnke laboratory.

Dr. Grogan's project involved applying highly specific immunoglobulin light chain reagents to cytospins of a number of lymphoma cell lines that were generated in Dr. Henry Kaplan's laboratory. There was tight control of reagents from the Kaplan laboratory with the bare minimum of cytospins provided after several requests, which usually involved detailed written justifications and some haggling. Dr. Grogan navigated these interactions with his customary Irish charm and diplomacy. The resulting manuscript included a few patient samples in addition to the Kaplan cell lines and heavy chain antibodies [11]. In this study of four Burkitt and five non-Burkitt lymphoma samples from children and adults, immunologic, cytochemical, ultrastructural, cytologic, histopathologic, clinical and cell

culture features were described. All samples stained for a single immunoglobulin light chain antibody indicating a monoclonal B-cell lymphoid proliferation and emphasized the significant overlap among the subtypes of "undifferentiated" lymphomas.

Another project that Dr. Grogan was involved in was an early application of a limited panel of mouse monoclonal antibodies to a collection of 30 frozen section samples of diffuse large cell lymphomas to help clarify their nature/origin with clinical correlation [12]. Anti-leu1 (CD5), anti-12E7 (CD99) and anti-HLA-DR, as well as immunoglobulin light and heavy chain reagents were applied. Dr. Grogan's experience with histochemistry for macrophages proved to be very helpful for this project. The resulting manuscript won the inaugural Benjamin Castleman Award and Dr. Warnke received the award from Dr. Castleman himself at the 1981 USCAP meeting.

## 4. The Promise of Immunohistochemistry

As the number of monoclonal antibodies increased, immunohistochemistry methods such as antigen retrieval and signal amplification techniques evolved as well. The use of biotin-avidin reagents and immunohistochemistry on frozen sections began to emerge [7]. In the Warnke laboratory, up to three frozen sections were placed on a single glass slide and stained with three different monoclonal antibodies. The biotin-avidin detection method also evolved such that instead of carefully washing, wiping and putting detection reagents on a glass slide, Coplin jars that contained the detection reagents were used for emersion of the slides containing the patient sample. These steps simplified the technical work by placing primary antibodies on slides on a laboratory bench followed by placing multiple slides in jars containing the detection reagents [13]. Eventually, staining vats were used and the art was in knowing when to change the jars of detection reagents before the amount of antibody was sufficiently exhausted that they would no longer work. Well after Dr. Grogan departed for the University of Arizona, he and Dr. Warnke kept in touch and shared their parallel efforts at "automation".

Another doyen, Dr. David Mason at the John Radcliffe Hospital at Oxford University, was an inspiring influence in hematopathology and immunodiagnosis, particularly in the development of monoclonal antibodies. These reagents were freely shared among the mavens of the day, including Drs. Grogan and Warnke. The combination of novel antibody reagents and technical advances in immunophenotyping, augmented both diagnostics and research through numerous highly productive collaborations. Investigations ranged from understanding the immune architecture of lymphoid organs, lineage determination, disease discovery and subtype classification of lymphomas, leukemias and histiocytic neoplasms. These avenues of research opened additional lines of inquiry and had a tremendous impact on the growth of hematopathology as a field [14–28]. A few significant examples that were contributed by Dr. Grogan in this time period are described below.

Using a battery of monoclonal antibodies specific to B-cells, T-cells and histiocytes, Dr. Grogan and colleagues characterized the immunoarchitecture of lymphoid organs such as the spleen and thymus [16,17]. These studies were instrumental in defining different compartments and cell types within those lymphoid organs. They provided a topographical framework for lymphomagenesis and helped refine diagnostic criteria for various lymphoma subtypes arising from the compartment and cell types from those organs.

Dr. Grogan was involved in large-scale studies to profile immunophenotypic features of Hodgkin and non-Hodgkin lymphomas. Inter-institutional comparative studies were used to provide systematic testing of the robustness of immunohistochemistry in different laboratories and establish early validation practices for reagents as well as methodology [18]. Variation in immunophenotypes and their impact on clinical behavior was exemplified by studies on peripheral T-cells lymphomas and lymphomas and myelomas involving the bone marrow [19,20]. These studies provided a window into the complexity of lymphoma markers as well as interpretation of the tumor cell phenotypes in the background of immune infiltrates forming the tumor microenvironment (TME).

In numerous examples among his publications, Dr. Grogan has shown how exceptions can become a driver for further study of underlying biologic and clinical differences. One such example is provided in his paper regarding lymphoproliferative disorders arising in acquired immunodeficiency syndrome (AIDS). The recognition that lymphoproliferative disorders in immunodeficiency states harbor distinct differences from those arising in patients with an intact immune system was recognized in the post-transplant setting. An astute observation by Dr. Grogan and colleagues showed that there were similar lymphoproliferative disorders in AIDS patients [21]. Initial biopsies of two patients with AIDS were found to have multifocal clusters of large blastic lymphoid cells; some clusters expressed a predominance of lambda light chains whereas others expressed kappa light chains. In subsequent biopsies, the patients developed frank lymphomas. That AIDS-associated lymphomas may arise from polyclonal lymphoproliferations followed by clonal evolution was a salient contribution to the field at a time when AIDS-associated neoplasms were beginning to be understood.

Another important attribute of Dr. Grogan's work is his keen interest in correlating histologic and immunohistologic observations with clinical features. Several examples can be found in the use of immunohistochemical markers to identify clinical behavior and risk groups. Dr. Grogan's study of the proliferation-associated antigen, Ki-67, revealed differences in growth fraction and clinical outcome among patients with diffuse large cell lymphoma (DLCL) [22]. Similarly, using a monoclonal antibody to the human class II histocompatibility antigen HLA-DR (Ia), the presence or absence of this antigen was found to correlate with outcome in DLCL [23]. The prognostic significance of the immunophenotype to distinguish among B- and T-cell derived non-Hodgkin lymphomas was clearly illustrated in another publication that highlighted differences associated with lineage determination in lymphomas [24]. Differences in clinical outcomes also led to explorations of mechanisms that underlie drug resistance such as P-glycoprotein [25,26]. His long-term interest in histiocytic and dendritic cell-derived neoplasms allowed him to make important contributions, along with members of the International Lymphoma Study Group, on the use of immunohistochemistry in the classification and subtyping of histiocytic and dendritic cell tumors [27,28]. These seminal contributions laid the foundation for the World Health Organization classification on the subject.

## 5. Realizing a Vision

Dr. Grogan credits the impetus for wanting to invent a fully automated immunohistochemistry instrument to his technologist, Ms. Catherine Rangel, and to the great and constantly growing need to deliver accurate and timely results on patients' tumor biopsies [1]. In 1985, immunohistochemistry was performed manually on single glass slides, involved 40–50 sequential steps and took many hours to complete. Comparison of stained slides between laboratories or even within the same laboratory from day-to-day was problematic with a failure rate as high as 10 percent. There were no standardized protocols. A scalable and efficient solution became imperative as every pathologist began to face the same need. Oncologists increasingly began to rely on highly honed pathology diagnoses annotated with immunohistochemistry to choose relevant treatments for their patients.

The need and the urgency to invent an automated platform were acute and the result was Ventana Medical Systems, Inc (Tucson, AZ, USA). Automated immunohistochemistry was piloted by a couple of academic centers including Stanford, adopting the early technology with the automated Ventana instrument in 1991. That trend rapidly increased as more and more medical centers chose automation to improve diagnostic accuracy and benefit from the efficient and scalable model that enabled timely diagnoses for all patients. By 2005, there were more than 5000 automated devices in 1500 institutions in 55 countries [1]. Automated platforms for immunohistochemistry continued to proliferate and become an integral part of academic and commercial laboratories. Its accessibility and ease of use that allowed standardization of assays and a high throughput of patient samples as part of the

daily workflow in the clinical setting was of great appeal. In academic immunodiagnosis laboratories today, it is not unusual to find up to 10 automated immunostainers and a menu of 250 markers that target a variety of tumor types and infectious agents. In 2008, after more than 20 years of working to revolutionize cancer diagnostics, Ventana Medical Systems, Inc. was acquired by Roche Diagnostics and incorporated as Roche Tissue Diagnostics. Roche Tissue Diagnostics continues to be at the forefront of precision diagnostics and innovation in anatomic pathology today.

In hematopathology, the impact of automated immunohistochemistry made it possible to routinely perform lineage determination and subtype classification with the immediate refinement of diagnoses. The Revised European American Lymphoma (REAL) classification released in 1995 [29], as well as the World Health Organization (WHO) classifications beginning from its first iteration in 2001 [30], followed by 2008 [31], and 2017 [32], were founded on the overarching concept that disease diagnoses must be based on a combination of clinical, morphologic, immunophenotypic and genetic features. This conceptual framework was largely aided by significant milestones in automated immunohistochemistry where clonality assessments, lineage determination, measurement of oncogene expression and predictive markers could all be taken into consideration and routinely evaluated to arrive at a final diagnosis.

Measurement of specific disease-defining markers by immunohistochemistry offered the opportunity to refine diagnostic criteria and bring uniformity to classification systems such that pathologists the world over could apply the same criteria and adhere to disease definitions. This first step in standardization also provided data-driven and robust comparisons among laboratories, which led to prolific scientific research, collaboration, and consensus guidelines in the field. Two specific disease-defining markers that became assessable by immunohistochemistry are exemplified by cyclin D1 (BCL1) expression in mantle cell lymphoma, and anaplastic lymphoma kinase (ALK) expression in ALK+ anaplastic large cell lymphoma (ALCL).

Lymphomagenesis is well known to be a multifactorial process where causality is often difficult to determine. In addition to oncogenic drivers, a subset of cancers including lymphomas is associated with infectious agents, particularly, pathogenic viruses. The ability to reliably detect viruses that are associated with lymphomas and other cancers, was another important step forward in defining disease processes and causative agents. Endemic Burkitt lymphoma from which the Epstein–Barr virus (EBV) was characterized in 1964 is a well-defined example [33]. The Ventana platform established an in situ hybridization (ISH) assay for the detection of EBV-encoded small RNAs (EBER). The EBER ISH assay was transformative in defining EBV-associated lymphoproliferative disorders. This technology allowed the detection of the virus directly in tissue and led to the study of cell and tissue types infected by the virus, latency patterns, and their contribution to lymphomagenesis. These studies resulted in the development of diagnostic criteria for the classification of EBV-associated lymphoproliferative disorders, particularly in the post-transplant setting, and in HIV-associated lymphomas [29–33]. KSHV/HHV8 is another virus whose latency-associated nuclear antigen (LANA)1 can be detected by immunohistochemistry in tissue biopsies. Detection of LANA1 aids in the classification of a spectrum of HHV8-associated lymphoproliferative disorders including multicentric Castleman disease, primary effusion lymphoma and HHV8+ DLBCL [29–32].

Perhaps the most compelling Ventana story of infectious agent-associated lymphoma diagnostics comes from the example of *Helicobacter pylori (H. pylori)*-induced gastric marginal zone lymphoma. A friend and colleague, Dr. Peter Isaacson at the University College London made the seminal observation that inflammation and ulcer-formation caused by *H.pylori,* if untreated over a prolonged period of time could evolve into a lymphoma of mucosa-associated lymphoid tissue or marginal zone lymphoma. Furthermore, this lymphoma could respond to antibiotic therapy targeting the bacterial infection. Upon first hearing this unprecedented story in 1992, Dr. Grogan realized the similarity to a patient at his home institution, who also subsequently responded to antibiotics [1]. Inspired by the

possibility that detection of *H. pylori* in gastric biopsies could prevent the development of cancer, he turned inspiration into action. Ventana developed an immunodiagnostic assay for *H. pylori*, which today remains one of its most frequently used assays worldwide.

### 6. Enabling Precision Health

Once immunohistochemical markers became available to identify different risk groups, interest in harnessing those differences to define therapeutic targets became possible. Additional increments followed from these initial discoveries, most importantly, the ability to measure the expression of treatment targets and refine patient-specific management plans, thereby ushering in an era of precision health. Ventana, under the innovative leadership of Dr. Grogan, was a highly influential enabler of the precision health vision by forging new technology, building efficient platforms and discovering and producing novel reagents. The resulting impact in the field of hematopathology is traced through several examples below.

Pioneering work in monoclonal anti-CD20 antibody (Rituximab) therapy in the 1990s and early 2000s provided the first example of chimeric antibody-based therapies for B-cell lymphomas [34–38]. Since its approval in 1994 for the treatment of B-cell non-Hodgkin lymphoma, rituximab has become a transformative agent in the clinical management of all types of low-grade and aggressive B-cell lymphomas. An example of the use of gene expression profiling to predict clinical outcome in DLBCL treated with RCHOP is provided in a study by Drs. Grogan and colleagues [39]. Today, rituximab used in frontline, relapse, maintenance and salvage regimens of B-cell non-Hodgkin lymphomas as well as in rheumatologic and autoimmune conditions [40,41]. The efficacy of targeted therapy relies on the measurement of target expression on lymphoma cells. The vast majority of B-cell lymphomas express CD20, which is routinely assessed in the immunohistochemical workup and diagnosis of B-cell lymphomas. The presence of CD20 approves a patient for rituximab-based therapies; measuring target downregulation in post-therapy settings and in recurrent disease are important endpoints in predicting duration and continued response to anti-CD20 therapy as well as overall clinical outcomes. Furthermore, the absence of CD20 in a newly diagnosed B-cell lymphoma is equally important to recognize such that alternative clinical management approaches can be pursued. The ability to measure CD20 expression in routine tissue biopsies suspected of lymphomas, is therefore vital in determining therapy.

Typically, the workup of a suspected lymphoma today involves an initial histologic evaluation followed by the application of an astutely chosen panel of immunohistochemical markers on an automated platform. This approach provides an immunophenotypic profile of the lymphoma, which achieves several objectives including lineage determination, subtype designation, evaluation of risk factors and predictive markers as well as therapeutic targets. In today's practice, this step may be further annotated by flow cytometry, cytogenetics including fluorescent in situ hybridization (FISH), co-occurrence of viruses such as EBV, as well as molecular clonality and pathogenic mutations by next-generation sequencing (NGS). An example of the workup and diagnosis of diffuse large B-cell lymphoma (DLBCL) is provided in Figure 2. This schematic shows the ancillary testing modalities most frequently used to obtain the relevant information and how these are combined in making a final WHO diagnosis.

Immunohistochemistry is currently used to determine the expression of several other therapeutic targets in lymphoma as well as other tumors. Brentuximab vedotin, an anti-CD30 antibody–drug conjugate is an example of targeted therapy that is particularly effective in its delivery of a cytotoxic drug to malignant cells bearing CD30. It is used in the treatment of classic Hodgkin lymphoma, anaplastic large cell lymphoma and other T-cell lymphomas including cutaneous T-cells lymphomas with CD30 expression [42–46]. The measurement of CD30 on neoplastic cells is performed by immunohistochemistry, which was found to be a valuable and precise tool to assess the expression of CD30, thereby enabling therapeutic decision-making [47,48].

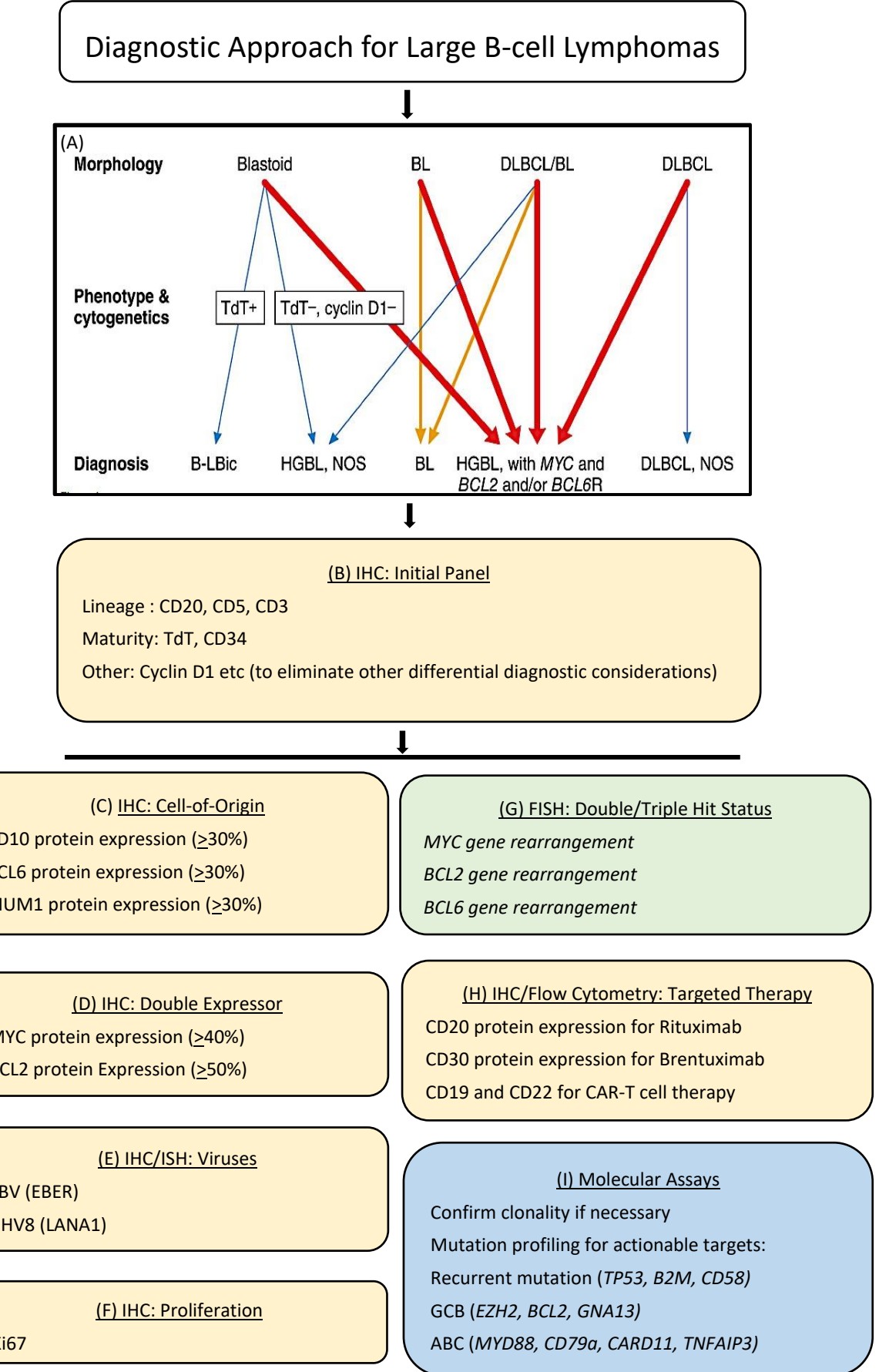

**Figure 2.** Precision diagnostic algorithm for diffuse large B-cell lymphoma. The diagnosis and subclassification of diffuse

large B-cell lymphoma today is highly annotated with ancillary studies, including many that rely on immunohistochemistry and in situ hybridization, which can be performed on the Roche/Ventana platform (shown in yellow). (**A**) Diagnostic algorithm for DLBCL according to WHO 2017 [32]. (**B**) Initial immunohistochemistry panel applied to tissue biopsies of atypical large cell proliferations; (**C**) Immunohistochemical algorithm based on cell-of-origin sub classification into germinal center (GC) versus non-germinal center (NGC) derivation; (**D**) Immunohistochemistry is used to detect MYC and BCL2 protein expression, which when co-expressed above predesignated cutoffs, identifies a DLBCL of intermediate prognosis; (**E**) Assays for the detection of EBV and HHV8, provide information regarding associated viruses and immune health; (**F**) Immunohistochemistry for Ki67 provides information regarding the proliferative capacity of lymphoma cells; (**G**) Fluorescence in situ hybridization (FISH) for *MYC, BCL2* and *BCL6* gene rearrangements provide information for the designation of double/triple hit lymphomas (*MYC* and *BCL2* and/or *BCL6* gene rearrangements), which designate more aggressive behavior and the need for potential escalation of therapy (shown in green); (**H**) immunohistochemistry as well as flow cytometry can be used to measure several targets within lymphoma cells, the expression of which defines eligibility for immune and cellular therapies; (**I**) mutational profiling by next generation sequencing provides important information regarding clinical behavior and actionable therapeutic targets (shown in blue).

Immune checkpoint blockade has recently emerged as an elegant strategy for targeted therapy in lymphoma as well as solid tumors and melanoma [49]. One such example is the blockade of programmed death 1(PD1), which restores immune function especially as it relates to the tumor microenvironment. Classic Hodgkin lymphoma is the prototype: the vast majority of cases harbor chromosomal alterations at *9p24.1* leading to overexpression of program death-1 (PD1) ligands, PD-L1 and PD-L2, and provide the substrate for anti-PD1-based therapies [50,51]. Anti-PDL1 immunohistochemistry is widely used in the measurement of various types of malignancies to evaluate the eligibility for anti-PD1-base checkpoint blockade therapy. The resounding success of this treatment strategy for various cancer types in the past five years has led to a sharp incline in competing antibody clones and immunohistochemical assays for the precise measurement of PD-L1. Ventana has been a key player in the early release of PD-L1 antibodies and assays to facilitate precision diagnostics related to immune checkpoint therapy. As exemplified in recent meta-analyses of PD-L1 immunohistochemistry [52,53], the complexity involved in the immunohistochemical assays for targeted therapies underscores the value and relevance of this technology in furthering the vision of precision diagnostics for many cancer types.

Genetically modified T-cells such as chimeric antigen receptor (CAR) T-cells, bi-specific T-cell engagers (BiTEs) and related cellular therapies are emerging as the next generation of lymphoma therapies. These strategies are increasingly requiring precise measurement including expression, downregulation and subcellular localization of several targets on lymphoma and leukemia tissue samples. Immunohistochemical assays, including those performed on the Ventana platform, as well as flow cytometry are routinely used to evaluate eligibility for clinical trials that are currently on-going for DLBCL, B-lymphoblastic lymphoma/leukemia, mantle cell lymphoma and plasma cell myeloma among others [54–57].

The mutational landscape of cancers can be reliably profiled by NGS applications [58]. These allow specific genetic alterations that are amenable for targeted therapy to be identified in a patient-specific manner. The cost and technical complexity of NGS, however, limits its widespread clinical applicability. A cost-effective alternative to molecular testing comes from mutation-specific antibodies that can facilitate the detection of a specific mutation by immunohistochemistry. One such example is *BRAF V600E*, a mutation in the *BRAF*-encoded serine/threonine kinase that is found in a variety of malignancies including melanoma, colorectal carcinoma, thyroid carcinoma, hairy cell leukemia, and Langerhans cell histiocytosis among other neoplasms [59,60]. Comparative DNA sequence analysis along with *BRAF V600E*-specific immunohistochemistry showed a high degree of concordance and supports the use of immunohistochemistry for the detection of this mutation in relevant cancer types [60]. In addition, antibodies that detect epigenetic modifications that are associated with hyper- or hypomethylation such as the H3K27me3 antibody allows the detection of histone H3 methylation at K27 in tissue biopsies. These methylation changes

are associated with *EZH2* mutational status and have been shown to confer prognostic differences in lymphomas and gliomas [61–63]. The H3K27me3 antibody assay is another example of how low-cost immunohistochemistry can circumvent the need for expensive molecular NGS assays to detect underlying disease mechanisms and prompt appropriate clinical management.

## 7. Conclusions

Dr. Grogan, through his highly innovative lens, has shepherded Ventana Medical Systems Inc., its affiliate organizations within Roche Diagnostics, and the broader field of pathology, on a quest to improve cancer diagnosis for all patients. Along his remarkable journey from the basement to co-founding a company valued in the billions, Dr. Grogan's vision for patient care remained crystal clear. He wanted to enable pathology diagnoses with improved accuracy and provide results in a timely manner to every cancer patient. Elevating the role and impact of a pathologist was intimately tied to that journey. For hematopathology, his innovation provided a foundation for disease discovery, classification and refinement of diagnostic criteria. They became fundamental to the basic tenets of the Revised European American Lymphoma (REAL) and World Health Organization (WHO) classifications of hematopoietic tumors that define hematopathology diagnoses. His journey continues today as Roche Diagnostics breaks new ground in the realm of multiplexed immunophenotyping, liquid biopsy, and digital–spatial profiling. The next steps of this journey and the future of Dr. Grogan's defining legacy of patient-centric innovation remain as bright and promising as ever.

**Author Contributions:** Both authors contributed to the writing. All authors have read and agreed to the published version of the manuscript.

**Funding:** This article received no external funding.

**Institutional Review Board Statement:** Not applicable.

**Informed Consent Statement:** Not applicable.

**Conflicts of Interest:** The authors declare no conflict of interest.

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
