# Peer review of "On the Shoulders of a Giant: Contributions of Thomas Grogan, MD to Hematopathology"

_hemato, doi:10.3390/hemato2010006_

Round 1

Reviewer 1 Report

A concise and informative review of the contributions of Dr. Thomas Grogan. I do not have any concerns except for the following minor ones.

Line 64, “entrée” needs to be “entry”.

Line 241, “and 2016 [32]”. The official publish year of the latest WHO classification is 2017.

Line 255, “anaplastic lymphoma kinase (ALK)-1 expression in ALK+ anaplastic large cell lymphoma (ALCL)”. ALK-1 is the clone name of a standard monoclonal antibody for ALK, and thus is not expressed by ALK+ ALCL, which expresses ALK.

Lines 271-272, “LANA” needs to be modified as “LANA1”.

Line 377 “limits is widespread clinical applicability” may be “limits its widespread clinical applicability”.

Author Response

Reviewer 1

A concise and informative review of the contributions of Dr. Thomas Grogan. I do not have any concerns except for the following minor ones.

  1. Line 64, “entrée” needs to be “entry”.
  2. Line 241, “and 2016 [32]”. The official publish year of the latest WHO classification is 2017.
  3. Line 255, “anaplastic lymphoma kinase (ALK)-1 expression in ALK+ anaplastic large cell lymphoma (ALCL)”. ALK-1 is the clone name of a standard monoclonal antibody for ALK, and thus is not expressed by ALK+ ALCL, which expresses ALK.
  4. Lines 271-272, “LANA” needs to be modified as “LANA1”.
  5. Line 377 “limits is widespread clinical applicability” may be “limits its widespread clinical applicability”.

Response: Thank you for the careful reading of our manuscript. We have made all of the suggested changes.

Reviewer 2 Report

The work is very well written and interestingly covers the career development and life of Dr. Grogan. 

However, minor clarifications and extensions are needed before the work is accepted for publication. 

pg 2. line 47

'by automated immunohistochemistry' - adding a paragraph already here, describing an early method/equipment in use would be beneficial.

pg 2.  line 72 - 73, 75 - 76 

Some clarification is needed for the following sentence on teaching adventure and the sentence after.

'They had a special affinity for each other and eventually traveled and taught together, most notably in a cross-country teaching adventure in Australia.' 

'Drs. Berard and Dorfman also shared a love of golf and many an A game would translate into impactful ideas that shaped the future of hematopathology.'

Figure 1 is missing.

Figure 2 - rearranging the information and adding colourous to how to diagnose the subtypes (GCB, ABC, PMBL) of DLBCL would make the figure more readable. 

The contribution of Grogan T. to rituximab-mediated therapy should be highlighted more clear - pg. 6. Authors should refer to the following work:

Gene expression predicts overall survival in paraffin-embedded tissues of diffuse large B-cell lymphoma treated with R-CHOP.

Rimsza LM, Leblanc ML, Unger JM, Miller TP, Grogan TM, Persky DO, Martel RR, Sabalos CM, Seligmann B, Braziel RM, Campo E, Rosenwald A, Connors JM, Sehn LH, Johnson N, Gascoyne RD.Blood. 2008 Oct 15;112(8):3425-33. doi: 10.1182/blood-2008-02-137372. Epub 2008 Jun 10.   Also, the contribution to of T. Grogan to the field of PD-L1 inhibition and CAR-T is not too clear. If it is through the work of Ventana, this must be clearly stated.  

Author Response

Reviewer 2

The work is very well written and interestingly covers the career development and life of Dr. Grogan. 

However, minor clarifications and extensions are needed before the work is accepted for publication. 

  1. pg 2. line 47: 'by automated immunohistochemistry' - adding a paragraph already here, describing an early method/equipment in use would be beneficial.

Response: Line 47 is part of the introduction. We did not want to elaborate on automated immunohistochemistry here but have done so in greater detail in sections 3, 4 and 5.

  1. pg 2.  line 72 - 73, 75 – 76: Some clarification is needed for the following sentence on teaching adventure and the sentence after. They had a special affinity for each other and eventually traveled and taught together, most notably in a cross-country teaching adventure in Australia.' 

'Drs. Berard and Dorfman also shared a love of golf and many an A game would translate into impactful ideas that shaped the future of hematopathology.'

Response: We have rephrased these sentences to clarify our meaning.

  1. Figure 1 is missing.

Response: We have included Figure 1 and the corresponding legend.

  1. Figure 2 - rearranging the information and adding colours as to how to diagnose the subtypes (GCB, ABC, PMBL) of DLBCL would make the figure more readable.

Response: Thank you for the suggestion. We have included colors to indicate the steps performed by immunohistochemistry or in situ hybridization using the Ventana/Roche platform. We have re-written the figure legend to clarify this point.

  1. The contribution of Grogan T. to rituximab-mediated therapy should be highlighted more clear - pg. 6. Authors should refer to the following work: Gene expression predicts overall survival in paraffin-embedded tissues of diffuse large B-cell lymphoma treated with R-CHOP.

Rimsza LM, Leblanc ML, Unger JM, Miller TP, Grogan TM, Persky DO, Martel RR, Sabalos CM, Seligmann B, Braziel RM, Campo E, Rosenwald A, Connors JM, Sehn LH, Johnson N, Gascoyne RD.Blood. 2008 Oct 15;112(8):3425-33. doi: 10.1182/blood-2008-02-137372. Epub 2008 Jun 10.  

Response: Thank you for pointing out this omission. We have added a sentence to the text (lines 299 – 301) and included the suggested reference [39].

  1. Also, the contribution to of T. Grogan to the field of PD-L1 inhibition and CAR-T is not too clear. If it is through the work of Ventana, this must be clearly stated.  

Response: We have clarified Ventana’s role in both PD-L1 (lines 360-342) and CAR-T cell therapy (lines 370-371).

Reviewer 3 Report

This is an outstanding manuscript written by world-renowned experts in the fields of pathology, immunohistochemistry, and hematopathology. It is a befitting piece about a pioneering 20th century figure whose innovative inventions in the field of immunohistochemistry ushered a new era of diagnostic and predictive pathology. The manuscript is written elegantly.  This reviewer has no substantive edits. 

Author Response

Reviewer 3

This is an outstanding manuscript written by world-renowned experts in the fields of pathology, immunohistochemistry, and hematopathology. It is a befitting piece about a pioneering 20th century figure whose innovative inventions in the field of immunohistochemistry ushered a new era of diagnostic and predictive pathology. The manuscript is written elegantly.  This reviewer has no substantive edits. 

Response: Thank you.